**Data Availability Statement:** All relevant data are within the manuscript and its supporting information files.

**Funding:** Author PR received National Heart, Lung, and Blood Institute Grant 2T35HL007649-36 to

# Billing rules / global period affect postoperative follow-up practices following total hip arthroplasty

Philip P. Ratnasamy⊚, Oghenewoma P. Oghenesume⊚, Peter Y. Joo⊚, Jonathan N. Grauer⊚*⊚

Department of Orthopedics & Rehabilitation, Yale School of Medicine, New Haven, Connecticut, United States of America

⊚ These authors contributed equally to this work.
* jonathan.grauer@yale.edu

## Abstract

### Background

Total hip arthroplasty (THA) is a common procedure following which postoperative visits are important to optimize outcomes. The associated global billing period includes the 90 postoperative days (or approximately 13 weeks), during which professional billing is included with the surgery itself. The current study assessed clinical practice patterns relative to the global billing period.

### Methods

Using the PearlDiver M91Ortho dataset, the incidence and timing of Evaluation and Management (E&M) codes in the 26 weeks following THA were assessed. The follow-up visits within and beyond the global billing period, and those conducted by surgeons versus non-surgeon advanced practice providers (APPs) were determined.

### Results

77,843 THAs were identified. Follow-up visits peaked at postoperative weeks 3, 7, and 14. The greatest week-to-week variation in the number of follow-ups was from weeks 13 to 14 immediately following the global billing period (representing a greater than 4-fold increase in visits.) During the first 13 postop weeks, 73.8% of patients were seen by orthopedic surgeons (as opposed to APPs). In the following 13 weeks, a significantly greater percentage of visits were with surgeons (86.8%, p<0.0001).

### Conclusions

Following the THA global billing period, there was marked increase in the number of follow-ups and transition to a greater percentage being performed by the surgeons. These results provide interesting insight into the potential impact of the billing structure on how practice is pursued.

support the completion of this study (https://www.nih.gov/). The funders had no role in study design, data collection and analysis, decision to publish, or preparation of the manuscript.

**Competing interests:** The authors have declared that no competing interests exist.

## Introduction

Total hip arthroplasty (THA) is a common orthopaedic procedure that can greatly improve the quality of life for affected individuals [1–3]. Postoperative visits are an important way for providers to track patient progress and provide ongoing care / counseling. It is hypothesized that billing structure and systems may affect the timing and nature of such visits.

The "global" billing period associated with the procedure such as THA is 90 days (or approximately 13 weeks) following the day of surgery, during which professional billing is included with the surgery itself. This is based on uniform rules regarding peri-operative reimbursement protocols developed by US Centers for Medicare & Medicaid Services (CMS) in 1992 [4–7]. Global billing regulations were enacted to ensure that payments made by Medicare Administrative Contractors (MACs) were consistent for similar services performed across varying jurisdictions, and to prevent excess Medicare payments for services more or less comprehensive than initially intended [8].

Some have argued that the "global" billing system does not account for the efforts of surgeons and their healthcare teams in caring for particularly ill patients during the global billing period. Providers may perform more work postoperatively without additional compensation might serve as a disincentive for surgeons to play active roles in the postoperative treatments within the initial 90 days following surgery. Ultimately, this may adversely affect patient care [5–7].

Despite the growing influence of billing protocols in modern medical practice, few studies have explored the potential impact of such protocols on clinical care. Thus, this retrospective database study aimed to analyze when, and by which types of providers (surgeons versus advanced practice providers [APPs]), THA patients are seen postoperatively in the context of the global and non-global billing periods.

## Materials and methods

### Study cohort

The PearlDiver M91Ortho dataset was used to abstract data for this retrospective cohort study. This is a large Health Insurance Portability and Accountability Act (HIPAA) compliant health administrative database containing aggregated and de-identified information on nearly 91 million orthopedic patients in the United States. Our Institutional Review Board (IRB) deemed studies using PearlDiver data exempt from review.

Inclusion criteria for the study included: Current Procedural Terminology (CPT) code 27130, which was used to identify patients who underwent total hip arthroplasty (THA). Further, only THA patients whose surgeons billed under the global billing period and those who had follow-up data available for at least the 26-week postoperative period were included. Exclusion criteria included: patients under the age of 18 and any patient with a diagnosis of hip fracture or infection (established by International Classifications of Disease codes [ICD]) on the day of their THA procedure.

### Postoperative follow-up visits

Postoperative visits during the 13-week THA global billing period were identified by postoperative CPT-99024. Postoperative visits during the following 13 weeks were identified by follow up CPT-99211, CPT-99212, CPT-99213, CPT-99214, and CPT-99215. Given the non-specific nature of the non-global period CPT codes, follow-up visits performed after the 13th postoperative week (i.e., during the non-global period) were only included in the present study if performed by a provider who billed for at least one global follow-up visit within the first 13 weeks

following THA. The number of postoperative follow-up visits conducted following THA was calculated by week through 26 weeks.

Providers performing the above-described follow up visits were grouped as surgeons and non-surgeons–with the non-surgeon group constituted of advanced practice providers (APPs) such as physician assistants and nurse practitioners. The proportion of follow-up visits conducted by surgeons and non-surgeons during the global and non-global billing periods was determined.

### Data analysis

The proportion of follow-up visits performed by surgeons during the global and non-global billing periods was compared by Z-test for two proportions, with statistical significance reached at $p < 0.05$. Prism9 (GraphPad Softwares, San Diego, CA) and Microsoft Excel (Microsoft Corporation, Redmond, WA) were used to create figures.

## Results

### Number of follow-ups during the 26 weeks following THA

Based on study inclusion/exclusion criteria, a total of 77,843 THA patients were identified, for whom 149,796 follow-up visits were recorded within the 26 weeks following THA. Of these visits, 62,658 (41.8%) occurred during the global billing period, compared to 87,138 (58.1%) during the subsequent non-global billing period (Table 1).

The incidence of postoperative follow-up visits by week is shown in Fig 1. Visits peaked at week 3 following THA at 10,904 (7.3% of all follow-up visits performed). The next peek was at 7 weeks following THA at 7,730 visits (5.2%). During the last week of the global billing period (week 13 following THA), 2,582 visits (1.72%) were performed, followed by a sharp rise to 10,738 visits (7.16%) in the first non-global billing week (week 14 following THA). The number of weekly follow-up visits then tended to decline through postoperative week 23, at which point 4,545 visits (3.0%) were performed. It subsequently rose to 6,714 visits (4.5%) in the 26[th] week following THA.

### Type of provider performing follow-up visit

The proportional distribution of provider types performing follow-up visits during the global and non-global billing periods is depicted in Fig 2. During the global billing period, 73.8% of follow-up visits were performed by surgeons, compared to 26.2% by non-surgeons. In contrast, during the following 13 weeks, 86.8% of follow-up visits were performed by surgeons, while non-surgeons performed 13.2%. A Z-test for two proportions found the difference in the proportion of follow-up visits conducted by surgeons during the global billing period compared to the non-global period to be statistically different ($p < 0.0001$).

## Discussion

The current study assessed treatment patterns of follow up visits and types of providers seen in the first and subsequent 13 weeks following THA to provide insight into the potential impact of modern billing practices on how clinical practice is pursued. There was marked increase in

**Table 1. Distribution of total hip arthroplasty follow-up visits during global and non-global billing periods.**

|  | Global Billing Period | Non-Global Billing Period |
|---|---|---|
| N (Total = 149,796) | 62,658 (41.8%) | 87,138 (58.1%) |

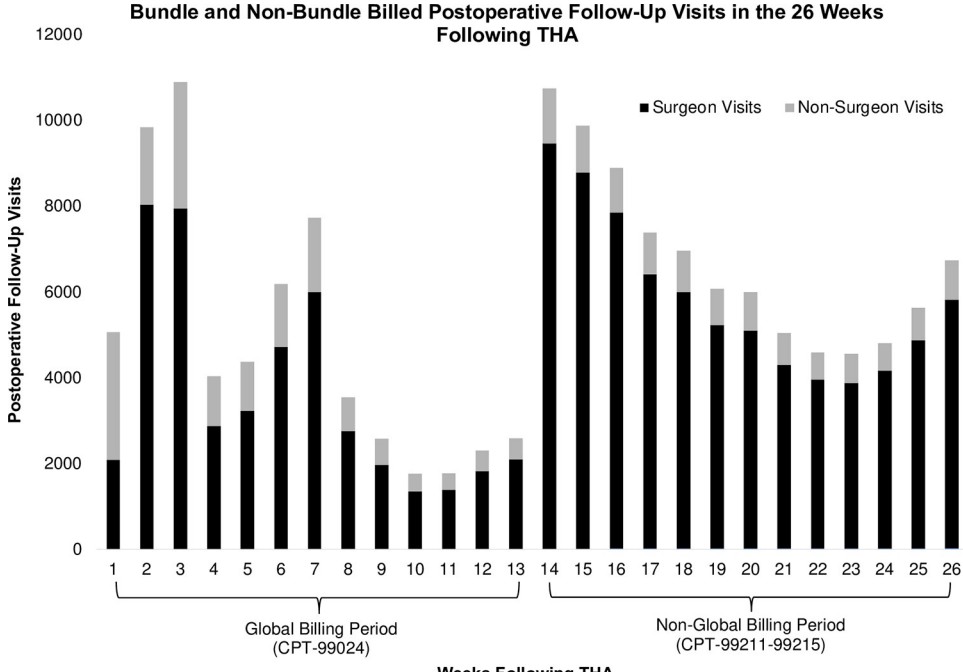

**Fig 1. Incidence of postoperative follow-up visits in the 26 weeks following total hip arthroplasty (THA), where weeks 1–13 constitute the global billing period and weeks 14–26 constitute the non-global billing period.** Follow-up visits were stratified as being performed by either surgeons or non-surgeons.

the number of follow-ups and transition to a greater percentage being performed by the surgeons following the global billing period.

In terms of THA postoperative follow up visits timing, 41.8% of visits were performed during the global period compared to 58.1% during the non-global period. This discrepancy is

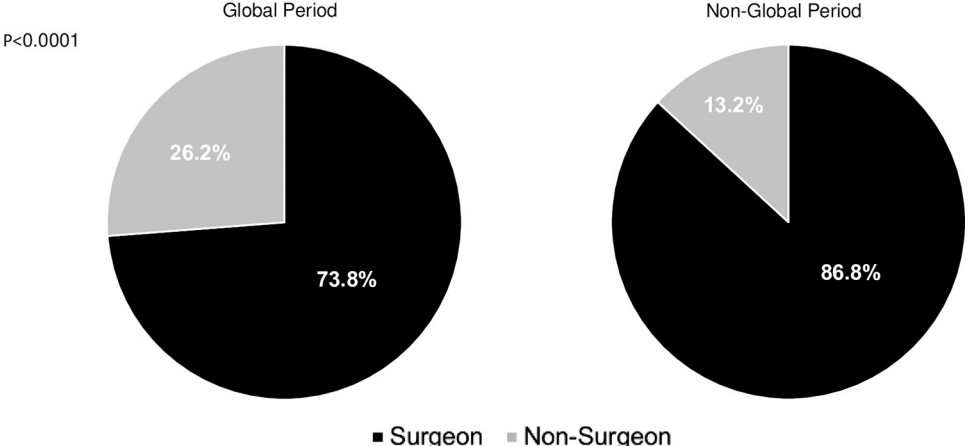

**Fig 2. Proportional distribution of THA follow-up visits conducted by surgeons and non-surgeons during the global billing period compared to the non-global billing period.** The difference in proportion of visits conducted by surgeons during the two billing intervals was highly statistically significant (p<0.0001).

noteworthy, particularly given that the first 13 postoperative weeks (corresponding to the global period) are the time of greatest expected adverse events and thus possible need for surveillance [9–11]. This finding aligns with a prior study by Curtis et al. who analyzed 381,561 nonoperative distal radius fractures billed with the nonoperative management code (that includes a global following period) versus evaluation and management codes (does not include a global following period) [12]. When disincentivized by a global billing structure, fewer office visits were performed within the 90-day global period (1.3 visits per patient during the global period vs. 2.3 visits during the non-global period).

The peaks in postoperative visits at week 3 (7.3% of all follow-up visits) and week 7 (5.2%) seem consistent with many clinical practices [13]. The largest transition in follow up timing was between weeks 13 (1.72%) and 14 (7.16%), corresponding to the transition between global and non-global billing. Furthermore, the percentage of patients seen by surgeons was lower during the global than non-global billing periods (73.8% vs. 86.8%). These findings suggest that the billing structure established by the global billing period may affect postoperative care timing and the types of providers from whom patients receive postoperative care.

As for limitations, the present study uses administrative data and is thus reliant and limited to the accuracy of the coded administrate data. The study only accounts for patients for whom their providers billed under the global billing period and for patients for whom we had consistent follow-up data for at least the 26-week postoperative period. Further, follow-up visits are not directly linked to index THA and thus could be for non-postoperative reasons; however, limiting follow-up visits to those performed by relevant orthopedic providers and the temporal proximity of visits to index THA should minimize the number of non-postoperative follow-up visits included in the present study. Additionally, it is possible that some patients were seen by both non-surgeon and surgeon providers during the global billing period, but this was not captured due to inaccurate billing/note signing. Despite this, the large variation in number of visits performed by non-surgeons and surgeons between the global and non-global periods highlights an overall trend in care and should negate individual billing errors.

To our knowledge, this is the first study assessing the effects, trends, and provider breakdown differences before and after the 90-day global billing period following a surgical operation. Importantly, the present study provides no insight into whether current billing practices negatively impact the quality-of-care patients receive, but rather solely seeks to explore whether billing protocols influence how clinical practice is pursued. These results suggest that billing protocols may influence care patterns, thus showing how administrative protocols impact care at the individual patient-provider level.

## Supporting information

**S1 Data.**
(XLSX)

## Author Contributions

**Conceptualization:** Philip P. Ratnasamy, Jonathan N. Grauer.

**Formal analysis:** Philip P. Ratnasamy.

**Investigation:** Philip P. Ratnasamy, Oghenewoma P. Oghenesume, Peter Y. Joo.

**Methodology:** Philip P. Ratnasamy, Oghenewoma P. Oghenesume, Peter Y. Joo, Jonathan N. Grauer.

**Supervision:** Jonathan N. Grauer.

**Writing – original draft:** Philip P. Ratnasamy, Oghenewoma P. Oghenesume, Peter Y. Joo.

**Writing – review & editing:** Philip P. Ratnasamy, Oghenewoma P. Oghenesume, Peter Y. Joo, Jonathan N. Grauer.

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
