## [Decision Letter · Decision Letter 0]

26 Dec 2023

PONE-D-23-26144Billing rules / global period affect postoperative follow-up practices

following total hip arthroplastyPLOS ONE

Dear Dr. Grauer,

Thank you for submitting your manuscript to PLOS ONE. After careful consideration, we feel that it has merit but does not fully meet PLOS ONE’s publication criteria as it currently stands. Therefore, we invite you to submit a revised version of the manuscript that addresses the points raised during the review process.

We look forward to receiving your revised manuscript.

Kind regards,

Stuart Barry Goodman, MD PhD

Academic Editor

PLOS ONE

Journal Requirements:

2. You indicated that ethical approval was not necessary for your study. We understand that the framework for ethical oversight requirements for studies of this type may differ depending on the setting and we would appreciate some further clarification regarding your research. Could you please provide further details on why your study is exempt from the need for approval and confirmation from your institutional review board or research ethics committee (e.g., in the form of a letter or email correspondence) that ethics review was not necessary for this study? Please include a copy of the correspondence as an ""Other"" file.

4. We note that your Data Availability Statement is currently as follows: All relevant data are within the manuscript and its supporting information files

Additional Editor Comments:

Please see the review and revise accordingly.

Reviewers' comments:

Reviewer's Responses to Questions

**Comments to the Author**

1. Is the manuscript technically sound, and do the data support the conclusions?

Reviewer #1: Yes

2. Has the statistical analysis been performed appropriately and rigorously? 

Reviewer #1: Yes

3. Have the authors made all data underlying the findings in their manuscript fully available?

Reviewer #1: Yes

4. Is the manuscript presented in an intelligible fashion and written in standard English?

Reviewer #1: Yes

5. Review Comments to the Author

Reviewer #1: Grauer et al. examine the influence of the "global" billing period on the timing and nature of postoperative follow-up visits following Total Hip Arthroplasty. The study analyzes data from PearlDiver database to investigate when and by which types of providers THA patients are seen in the context of global and non-global billing periods. This is an interesting study and adds to a less well studied area of patient care dalthough it mostly establishes an observation.

1. The study design does not allow for the establishment of causality and furthermore does not show this led to an increase in complications. If simple to do, looking at the same database and establishing if there were further complications requiring surgery for these patients by looking at procedure codes in the same timeline (or up to 2 years after) could potentially substantially add to the paper’s final discussion and conclusions and improve its clinical relevance.

2. Administrative data was used which is vulnerable to coding inaccuracies. Only looking at visits coded with 99212-5 are not enough as there are multiple other reasons patients could be seen other than post-operative visits, such as planned contralateral hip arthroplasty or total knee arthroplasty. Can visit be associated with diagnoses of visit to limit this cofounder? There is also the possibility that the patients were seen by midlevel and surgeons in conjunction during the billing period but were less diligent about signing the notes.

3. How missing data was managed could be expanded to better explain how these were addressed.

Possible grammatical/formatting errors noted

Line 42: “improve quality of life “–> “improve the quality of life“

Line 42: “individuals.(1, 2),(3)” –> “individuals.(1-3) “

Line 42: “for the similar services” –> “for similar services“

Line 143: “week 7 (5.2%) seems consistent” –> “week 7 (5.2%) seem consistent “

6. PLOS authors have the option to publish the peer review history of their article (what does this mean?). If published, this will include your full peer review and any attached files.

Reviewer #1: No

---

## [Author Response · Author response to Decision Letter 0]

7 Feb 2024

Thank you for taking the time to review our manuscript, “Billing rules / global period affect postoperative follow-up practices following total hip arthroplasty.” Please see responses to editor/reviewer queries below.

Response to Editor Comments: 

Response: We have updated file names appropriately, thank you.

2. You indicated that ethical approval was not necessary for your study. We understand that the framework for ethical oversight requirements for studies of this type may differ depending on the setting and we would appreciate some further clarification regarding your research. Could you please provide further details on why your study is exempt from the need for approval and confirmation from your institutional review board or research ethics committee (e.g., in the form of a letter or email correspondence) that ethics review was not necessary for this study? Please include a copy of the correspondence as an ""Other"" file

Response: Our IRB listed the following statement as reason for the present project being exempt from IRB review “The Yale IRB determined that the investigator is not engaged in research involving human subjects. As such, IRB review and approval are not required.” We have attached a copy of the exemption form from our IRB to the submission as well. Thank you.

Response: Thank you for this information.

4. We note that your Data Availability Statement is currently as follows: All relevant data are within the manuscript and its supporting information files. Please confirm at this time whether or not your submission contains all raw data required to replicate the results of your study. Authors must share the “minimal data set” for their submission. PLOS defines the minimal data set to consist of the data required to replicate all study findings reported in the article, as well as related metadata and methods. 

Response: We have attached an additional file containing raw data (supporting information file) used to build graphs for the present study. All data required to replicate study findings is present in the manuscript; however. 

Response: All references are accurate. None have been retracted. 

Response to Reviewer Comments: 

1. The study design does not allow for the establishment of causality and furthermore does not show this led to an increase in complications. If simple to do, looking at the same database and establishing if there were further complications requiring surgery for these patients by looking at procedure codes in the same timeline (or up to 2 years after) could potentially substantially add to the paper’s final discussion and conclusions and improve its clinical relevance.

Response: Thank you for this comment. Although the analysis you suggest could be interesting, our goal with the current manuscript is simply to characterize trends in the timing/provider type of follow-up visits and how this may relate to global billing practices. We do not aim to call into question the validity of current practices or suggest that current practices lead to inadequate patient care. 

2. Administrative data was used which is vulnerable to coding inaccuracies. Only looking at visits coded with 99212-5 are not enough as there are multiple other reasons patients could be seen other than post-operative visits, such as planned contralateral hip arthroplasty or total knee arthroplasty. Can visit be associated with diagnoses of visit to limit this cofounder? There is also the possibility that the patients were seen by midlevel and surgeons in conjunction during the billing period but were less diligent about signing the notes.

Response: Thank you for these valid points. Unfortunately, we are unable to characterize diagnoses associated with follow-up visits due to database limitations. In efforts to reduce confounding, we have only included follow-up visits by relevant orthopedic surgeons. The proximity to index hip arthroplasty may also limit the likelihood that follow-up visits are for alternative reasons. The point you raise regarding both mid-levels and surgeons seeing patients during visits is also valid and is unfortunately another limitation of the present study. Despite this, the large variation in number of visits performed by non-surgeons and surgeons between the global and non-global periods highlights an overall trend in care and should negate individual billing errors. We have now included both factors you mention in the limitations section of the manuscript. 

3. How missing data was managed could be expanded to better explain how these were addressed.

Response: We have further clarified our inclusion criteria in the methods of the study which only included patients whose surgeons billed under the global billing period and who follow-up data in the database for at least the 26-week postoperative period. Thus, all patients included in the final cohort had no missing data pertinent to the present study. 

Possible grammatical/formatting errors noted

Line 42: “improve quality of life “–> “improve the quality of life“

Line 42: “individuals.(1, 2),(3)” –> “individuals.(1-3) “

Line 42: “for the similar services” –> “for similar services“

Line 143: “week 7 (5.2%) seems consistent” –> “week 7 (5.2%) seem consistent “

Response: We have made these corrections in the manuscript. Thank you.

---

## [Decision Letter · Decision Letter 1]

28 Feb 2024

Billing rules / global period affect postoperative follow-up practices following total hip arthroplasty

PONE-D-23-26144R1

Dear Dr. Grauer,

We’re pleased to inform you that your manuscript has been judged scientifically suitable for publication and will be formally accepted for publication once it meets all outstanding technical requirements.

Kind regards,

Stuart Barry Goodman, MD PhD

Academic Editor

PLOS ONE

Additional Editor Comments (optional):

Thank you for revising your manuscript. This is an interesting, provocative study that should engender much discussion and controversy!

Reviewers' comments:

Reviewer's Responses to Questions

**Comments to the Author**

1. If the authors have adequately addressed your comments raised in a previous round of review and you feel that this manuscript is now acceptable for publication, you may indicate that here to bypass the “Comments to the Author” section, enter your conflict of interest statement in the “Confidential to Editor” section, and submit your "Accept" recommendation.

Reviewer #1: All comments have been addressed

2. Is the manuscript technically sound, and do the data support the conclusions?

Reviewer #1: Yes

3. Has the statistical analysis been performed appropriately and rigorously? 

Reviewer #1: Yes

4. Have the authors made all data underlying the findings in their manuscript fully available?

Reviewer #1: Yes

5. Is the manuscript presented in an intelligible fashion and written in standard English?

Reviewer #1: Yes

6. Review Comments to the Author

Reviewer #1: Grauer et al. examine the influence of the "global" billing period on the timing and nature of postoperative follow-up visits following Total Hip Arthroplasty. The study analyzes data from PearlDiver database to investigate when and by which types of providers THA patients are seen in the context of global and non-global billing periods. This is an interesting study and adds to a less well studied area of patient care which is becoming more relevant in today’s rapidly evolving healthcare stetting.

They have adequately addressed my previous comments.

7. PLOS authors have the option to publish the peer review history of their article (what does this mean?). If published, this will include your full peer review and any attached files.

Reviewer #1: No

---

## [Editor Report · Acceptance letter]

22 Mar 2024

PONE-D-23-26144R1 

PLOS ONE

Dear Dr. Grauer, 

I'm pleased to inform you that your manuscript has been deemed suitable for publication in PLOS ONE. Congratulations! Your manuscript is now being handed over to our production team.

Kind regards, 

on behalf of

Dr. Stuart Barry Goodman 

Academic Editor

PLOS ONE